



# DO-like events of the penultimate climate cycle: the loess point of view

Denis-Didier Rousseau[1,2], Pierre Antoine[3], Niklas Boers[4], France Lagroix[5], Michael Ghil[1, 6], Johanna Lomax[7], Markus Fuchs[7], Maxime Debret[8], Christine Hatté[9], Olivier Moine[3], Caroline Gauthier[9], Diana Jordanova[10], Neli Jordanova[10]

[1]Ecole Normale Supérieure, Laboratoire de Météorologie Dynamique, UMR8539 CNRS, Paris cedex 05, France
[2]Lamont Doherty Earth Observatory, Columbia University, New York, USA
[3]Laboratoire de Géographie Physique, Environnements quaternaires et actuels, CNRS, Meudon, France
[4]Potsdam Institute for Climate Impact Research, Potsdam, Germany
[5]Institut de Physique du Globe de Paris, Sorbonne Paris Cité, University Paris Diderot, UMR7154 CNRS, Paris, France
[6]Department of Atmospheric and Oceanic Sciences, University of California, Los Angeles, CA 90095-1565, USA
[7]Department of Geography, Justus-Liebig-Universität Gießen, Gießen, Germany
[8]Normandie University, UNIROUEN, UNICAEN, CNRS, M2C, Rouen, France
[9]Laboratoire des Sciences du Climat et de l'Environnement, Université Paris-Saclay, Gif-sur-Yvette, France
[10]National Institute of Geophysics, Geodesy and Geography, Bulgarian Academy of Sciences, Acad. G. Bonchev Str., Block 3, 1113, Sofia, Bulgaria

*Correspondence to*: Denis-Didier Rousseau (denis-didier.rousseau@lmd.ens.fr)

**Abstract**: The global character of the millennial-scale climate variability associated with the Dansgaard-Oeschger (DO) events in Greenland has been well-established for the last glacial cycle. Mainly due to the sparsity of reliable data, however, the spatial coherence of corresponding variability during the penultimate cycle is less clear. New investigations of European loess records from MIS 6 reveal the occurrence of alternating loess intervals and paleosols (incipient soil horizons), similar to those from the last climatic cycle. These paleosols are correlated based on their stratigraphical position and numbers, and available optically stimulated luminescence (OSL) dates with interstadials described in various Northern Hemisphere records as well as in GLt_syn, the synthetic 800-kyr record of Greenland ice core δ[18]O. Therefore, referring to the interstadials described in the record of the last climate cycle in European loess sequences, the MIS 6 interstadials can confidently be interpreted as DO-like events of the penultimate climate cycle. The statistical similarity between the millennial-scale loess-paleosol oscillations during the last and penultimate climate cycle provides direct empirical evidence that the cycles of the penultimate cycle are indeed of the same nature as the DO cycles originally discovered for the last glacial cycle. Our results thus imply that their underlying cause and global imprint was characteristic of at least the last two climate cycles.

## 1. Introduction

The last climate cycle (130-12 ka) has been punctuated by two different types of abrupt climate changes. The first one corresponds to abrupt warmings of up to 16.5°C (Kindler et al., 2013) in about 50 years over Greenland, followed by returns to glacial conditions in about several centuries. Described as Dansgaard-Oeschger (DO) events, 26 interstadials occurred over the last 115 -12 ka b2k interval at varying frequencies. A recent interpretation of their occurrence is that they result from the interactions between i) variations in the expansion of the ice shelves surrounding the three main ice sheets covering the northern hemisphere and more especially Greenland, ii) the expansion of the sea ice, and iii) alterations of the Atlantic Meridional Overturning Circulation ( Petersen et al., 2013; Boers et al., 2018). The DO events have also been described in variations of planktonic counts characterizing sea surface temperatures in North Atlantic cores recording the last climate cycle (Bond et al., 1992; McManus et al., 1994; Clark et al., 1999; Clark et al., 2007; Pisias et al., 2010).

The second type of abrupt climate change, called Heinrich events ( Heinrich, 1988; Broecker, 1994), is most likely related to the DO events and appeared at the end of several, but not all, DO cycles and named Bond cycles (Alley, 1998). The Heinrich events are characterized by massive iceberg discharges calved from the 3 main ice sheets into the North Atlantic. They are



identified in North Atlantic records by the occurrence of mineral detritus, eroded from North America, Greenland and Northern Scandinavia, carried out by the massive icebergs and deposited at the bottom of the ocean after the melt of the icebergs. These particular iceberg detritus deposits have been identified in the North Atlantic records down to 650 ka (MIS 16) from the Hudson strait ( Hodell et al., 2008; Hodell and Channell, 2016).

The Greenland ice-core warm intervals had been initially correlated with the classical terrestrial interstadials described in European pollen records (Dansgaard et al., 1993). Since then, other correlations have been proposed from numerous European pollen sequences. A more detailed overview of these terrestrial records shows that indeed the European pollen sequences, either terrestrial or marine, did record numerous warm and humid intervals during the last climate cycle, which are synchronous to the Greenland and North Atlantic interstadials (Fletcher et al., 2010). During the DO events, the temperate forest in southern latitudes opposed to the more open shrub-tundra grassland in northern latitudes, higher than 45°N.

More recently, the high-resolution study of the last climate cycle in European loess-paleosol sequences at about 50°N has demonstrated that the Greenland interstadials described in the Greenland ice-cores correspond one-to-one with the paleosols preserved in European loess (Rousseau et al., 2002; Rousseau et al., 2007; Rousseau et al., 2017a; Rousseau et al., 2017b). Contrary to the pollen records, where some dating uncertainty remains, the $^{14}$C dates obtained from earthworm granules preserved in each individual paleosol described in the loess sequences perfectly confirm this correlation (Moine et al., 2017). These new results confirmed the initial hypothesis that the length of the Greenland interstadials (GIS) corresponds to the matureness of the corresponding paleosols in Europe, from boreal brown paleosols, to tundra gleys, and even embryonic tundra gley horizons. Interestingly, this relationship between GIS and European paleosol-loess alternation is rather observed in sequences located at about 50°N, while less or not visible further southward and more difficult to evidence eastward where the environment was too dry for the formation of soils (Rousseau et al., 2011).

In the present paper, we discuss the MIS6 paleosol-loess succession recently released by the multiproxy high-resolution investigation of the very well developed loess-paleosol record of Harletz in Bulgaria (Antoine et al., 2019; Lomax et al., 2019) and compare this succession with other regional (Mediterranean and European) and more global records (Northern Hemisphere). The aim of this research is to understand if DO-like events also prevailed prior the last climate cycle, although there is no direct record of the penultimate climate cycle in Greenland.

## 2. Harletz paleosol succession (Fig. 1)

The Harletz loess sequence is a 20 m thick terrestrial record located in Bulgaria on the left bank of the Ogosta river, a tributary of the Danube River, at 43°41'52.78" N, 23°49'42.27" E, alt. + 40m a.s.l. (Fig. 1)

The multidisciplinary high-resolution investigation - including grain size, organic carbon (%), magnetic susceptibility and spectro-colorimetry - of the loess deposits and inter-stratified soil units has revealed the superposition of two interglacial-glacial cycles below the top soil (Antoine, 2019). Combined with luminescence dating (Lomax et al., 2019), this approach allowed to correlate these pedosedimentary cycles with loess cycles C and B of Kukla (1977), which are themselves correlated to Marine Isotope Stages (MIS) 7-6 and to MIS 5-2, respectively.

The lower cycle is composed of a 4 m thick basal interglacial pedocomplex (Ogasta soil complex) corresponding to an upbuilding soil sequence superimposed on overbank fluvial sandy silts from the Ogosta river overlain by a first eolian series, 10 m thick, showing a detailed record of alternating loess and paleosol units. The upper cycle starts with a 2 m thick interglacial paleosol (Harletz soil complex) overlain by a loess layer, about 4 m thick, capped by a modern-day truncated soil (20 cm) (Antoine et al. 2019). The sedimentological study shows that the two eolian units are built from regional wind-blown silts and fine sands. The general stratigraphy identified in Harletz fits with the general frame described for the nearby sequences from Serbia, Romania and Hungary by Markovic et al (2015), including the identification of a cryptotephra layer within cycle C, mainly identified through magnetic susceptibility but also with the occurrence of a few glass shards



(Sébastien Nomade, oral communication). Nevertheless, the identification of the paleosol succession at the base of the penultimate eolian sequence is particularly unique (Antoine et al., 2019) for European loess sequences. This soil sequence exhibits a succession of four soil horizons separated by so many loess deposits, in which the thickness and the intensity of each pedogenesis is less and less developed towards the top (Antoine et al., 2019). Moreover, while most of the paleosols (even some ≤ 10cm thick) have been observed when cleaning the outcrop, others are very incipient paleosols, incipient

weathered horizons, and could only be identified through magnetic susceptibility and spectrocolorimetry measurements supported by grain size variations (Fig. 1).

The luminescence dating obtained for the upper part of the stratigraphy above the cryptotephra layer are consistent with the nearby sequences and support our interpretation of the MIS 7-6 and MIS 5-2 assignment for the lower and the upper part of the section, respectively (Lomax et al., 2019). A tephra layer has also been described in other loess sequences of the region

(Lomax et al., 2019). However, no date has been directly measured on this particular marker, but all existing estimates indicate a minimum age of ca. -170 ka. (Antoine et al., 2019). This age can be compared with two volcanic eruptions, identified south of Rome/Italy, the Vrico B ignimbrite and the Pitigliano Tuff (Giraudi and Giaccio, 2017), but also in nearby Lake Ohrid, where the tephra have been dated to -162 ± 6 ka and -163 ± 22 ka, respectively (Leicher et al., 2016). In Harletz, the luminescence date of the sample 0.5 m above the cryptotephra indicates an age of -171 ± 14 ka, consistent with

estimates of the tephra layer from Mosticea to the east, and Rumac and Stalac to the northwest of Harletz. At Harletz, the lower record of MIS6 indicates 5 paleosols to incipient weathered horizons of various maturation (ISp). They are distributed along a decreasing trend in the clay content, in the color reflectance and the magnetic susceptibility, but also along an increasing trend in the coarse sand and in the grains size ratio (coarse + fine sand/clay) (Fig. 1).

Just above the cryptotephra layer, there are two more thin incipient weathered horizons only observed through the analytical

data (cryptopaleosols ISm). Moreover, ongoing magnetic studies (isothermal remanent magnetization and coercivity of remanence) identified 3 more incipient weathered horizons on top of the loess unit at 6.2m, 7.4m and 8.4m (Lagroix et al., 2016). Therefore, considering the Lake Ohrid recalculated dates of the tephra layer and the luminescence age available, there are 10 horizons including paleosols and incipient weathered horizons identified above the MIS7 Ogosta interglacial pedocomplex. In the MIS6 loess section, there are 7 and 3 horizons in the lower and the upper parts, respectively, assigned to

the interval between -190 and -130 ka. These paleosols and incipient horizons are interpreted to correspond to interstadials of various duration in the continental record by reference to those that have been described in the last climate cycle (Antoine et al., 2009; Rousseau et al., 2002; Rousseau et al., 2007; Rousseau et al., 2017a) (Fig. 1).

Although unique among the other loess sequences from the area, can these paleosols-incipient weathered horizons/interstadials be related to more global events as interpreted for last climate cycle paleosols?

**3. Comparison with closest Mediterranean records:**

**3.1 Loess records (Supp. Fig. 1),**

Numerous loess sequences within the Carpathian and Lower Danube Basins show the identified penultimate loess unit assigned to MIS 6 or early MIS 5e, either based on luminescence ages or on magnetic susceptibility variations, in Bulgaria ( Jordanova and Petersen, 1999; Jordanova et al., 2008), in Serbia (Fuchs et al., 2008; Markovic et al., 2009; Murray et al.,

2014), in Ukraine (Buggle et al., 2009), in Romania (Balescu et al., 2010; Timar-Gabor et al., 2011; Vasiliniuc et al., 2012) and in Hungary (Novothny et al., 2010; Novothny et al., 2011; Ujvari et al., 2014). Some of them show the occurrence of a tephra layer, e. g. in Batajnica, Ruma and Stalac in Serbia (Buggle et al., 2009; Markovic et al., 2009; Novothny et al., 2010; Novothny et al., 2011; Obreht et al., 2016; Ujvari et al., 2014; Vandenberghe et al., 2014), and at Mostistea in Romania (Balescu et al., 2010; Panaiotu et al., 2001). Markovic et al. (2015) did not assign this tephra to any particular eruption, but

considering the age model proposed, it rather corresponds to either the Vico B Ignimbrite or the Pitigliano Tuff (Giraudi and



Giaccio, 2017) at about -170 ka. Furthermore, in the review of Danube loess sequences, Markovic et al. (2015) also refers to an embryonic pedogenic layer, older than the identified -170 ka tephra but younger than the first loess subunit identified at the start of the L2 loess unit correlated with MIS6, above the interglacial paleosol S2SS1. Because of its position in the stratigraphy scheme, this embryonic paleosol is correlated with the first paleosol identified in Harletz between about -15 m

and -14.5 m depth (Antoine et al., 2019), Supp. Fig. 1). The other paleosols identified in Harletz have not been mentioned in the Carpathian and Lower Danube loess sequences, making this record a remarkable reference sequence (Antoine et al., 2019).

**3.2 Pollen records (Fig. 2, Tab. 1)**

Located at 40°54'-41°10' N and 20° 38'-20° 48'E in the Balkan Peninsula within the Dinaride–Albanide–Hellenide

mountain belt, Lake Ohrid is a transboundary (Macedonia and Albania) lake. This site is the closest pollen record to the Harletz sequence, where two different open vegetation zones represent MIS6, namely OD-5 (-190 to –160 ka) and OD-4 (-160 to –129 ka) (Sadori et al., 2016). The former corresponds to a grassland dominated environment and the latter to a rather steppe-dominated environment. A tephra layer marks also the boundary between the two vegetal formations, indicating that OD-5 and OD-4 could be correlated with the lower and the upper parts of the Harletz MIS6 record, respectively (Antoine et

al., 2019). Considering the variations in both the percentage of total arboreal (AP) or total arboreal minus (pine+juniper+birch) pollen grains in OD-5, only 4 tree expansion phases can be noticed, which could correspond to 4 out of the 5 interstadials identified in the lower part of the Harletz MIS6 record. Their onsets are dated, according to the age model used (Sadori et al., 2016), at about -185 ka, -178 ka, -175 ka and 169 ka, below the tephra layer. In the following OD-4, 4 other interstadial onsets are dated at about -159 ka, -151 ka, -145 ka and -139 ka.

At lower latitude, the Tenaghi Philippon site is located in Greece at 41°10'N, 24°20'E; 40 m a.s.l. This is a pollen record famous for its long continental climate record covering about 8 climate cycles (Tzedakis et al., 2006). This record indicates woody taxa expansions, which can be considered as interstadials, during the interval equivalent to MIS6. Following the age model used (Tzedakis et al., 2006), they are occurring at about -185 ka, -180 ka, -171 ka, -163 ka, -153 ka, -151 ka and -138 ka, respectively. Therefore, some of them seem potentially synchronous to the tree expansions described at higher resolution

in Lake Ohrid and by extension to Harletz interstadials. Another long pollen core retrieved in Greece, Ionnina I-284 core (39°45' N, 20°51' E, 472.69 m a.s.l.), which covers several climate cycles as well, reveals the record of MIS 6 in the section -138 m to -99 m. Roucoux et al. (2011) have described the high-resolution variations of the vegetation during this time interval. Phases of temperate tree expansion, interpreted as interstadials, and phases of tree contraction, interpreted as stadials, have been identified. Based on the employed age model (Roucoux et al., 2011), these temperate tree expansions

occurred at about -186 ka, -181 ka, -177 ka, -173 ka, -170 ka, -165 ka, -159 ka, -156 ka, -145 ka, -138 ka and -135 ka that we labeled I11 to I1 (Fig. 2). They appear more numerous than in Tenaghi Philippon and in Lake Ohrid, probably due to the higher resolution of the pollen analysis of Ionnina core and similarly some of these tree expansions could be correlated with the Harletz MIS6 interstadials, five expansions being recorded older than -165 ka (Tab. 1).

**3.3 Marine records (Fig. 2, Tab. 1)**

On both sides of the strait of Gibraltar, Martrat et al. (2004; 2007) have described the climate variations expressed during at least the past two climate cycles from cores retrieved from the Alboran (ODP 977A, 36°1.9'N, 1°57.3'W; 1984 m below sea level) and the Iberian Seas (MD01-2443, 37°52.85'N,10°10.57'W, 2925 m below sea level). The analysis of the $d^{18}O$ from planktonic foraminifera *Globorigena bulloides* shows variations with abrupt changes mimicking – for the last climate cycle (MIS 5 to 2) – those described in the Greenland ice core records. During the penultimate cycle (MIS 7-6, -245 to -130 ka)

the foraminifera $\delta^{18}O$ record also shows abrupt warmings similar to the DO interstadials, which have also been observed in





the variations of the Uk'37 alkenone index, a proxy for the sea surface temperature. Nine interstadials are therefore identified, named Alboran or Iberian Margin interstadials, although the older one, AI-9', seems to be a composite one, in which 3 different events could be discriminated. The onsets of these penultimate cycle interstadials are dated at about: -186 ka (-186 ka, -182 ka, -179 ka), -176 ka, -170 ka, -165 ka, -159 ka, -152 ka, -150 ka, -141 ka and -133 ka, respectively. They were labeled AI-9', AI-8', AI-7', AI-6', AI-5', AI-4', AI-3', AI-2', and AI-1', respectively. Similarities in terms of the number of warming events appear with the long Mediterranean pollen records mentioned previously. Although no tephra is identified, 5 events can be once more identified prior to -165 ka, as also observed in Harletz (Fig. 2, Tab. 1).

**3.4 Speleothem records (Fig. 2, Tab. 1)**

Two Mediterranean speleothems provide records for MIS6. The Argentarola Cave, located at (42°23'8.79"N 11°3'59.17"E) by the Tyrrhenian Sea, shows an interval of low $\delta^{18}$O values between 180 ka and 165 ka and interpreted as corresponding to the Mediterranean sapropel S6 (Bard et al., 2002) (Fig. 2). The recorded variations are correlated with total organic content variations recorded in the two Mediterranean cores MD84641 and KC19C, indicating rather warm and pluvial conditions during this interval over the western Mediterranean. Eastward, in Soreq cave in Israel, at 31.458 N 35.038 E and 400 m a.s.l, a speleothem record covering the last 180 ka was obtained. The measured $\delta^{18}$O record of MIS6 shows variations at about - 177 ka, -170 ka, -163 ka, -160 ka, -150 ka, -147 ka, -142 ka, -138 ka and -134 ka (Fig. 2). The peaks at -174 ka and -156 ka show very low values, interpreted as corresponding to intense pluvial intervals over Southern Israel, with the former correlated with Sapropel S6, while no particular sapropel is associated with the latter (Ayalon et al., 2002; Bar-Matthews et al., 2003). These records indicate high precipitation rates at least during the -180 ka to -165 ka interval, which also corresponds to the most clearly expressed interstadials/paleosols in Harletz.

Although the interstadials identified in the Harletz MIS6 sequence have not been described in the closest loess sections from the Carpathian Basin and lower Danube region, the overview of the various and diversified MIS6 records presented above clearly indicates that these paleosols and incipient weathered horizons correspond to climatic events that are recognized all along the Northern Mediterranean, and therefore have more than a local/regional significance. This shows that the climate mechanism proposed to explain the paleosol-loess alternations of the last climate cycle (Antoine et al., 2009; Boers et al., 2017;2018; Rousseau et al., 2002; 2007; 2017a; 2017b), in northern European sequences at about 50°N, seem to have prevailed already during the penultimate climate cycle. This suggests that these alternations are part of a more global dynamics. Therefore, in the next step of our study we will look for similar events in other records outside the Mediterranean region, at a more global scale.

**4 Paleoclimatology: DO-like events during MIS6 (Fig. 3, Tab. 1)**

The present investigation of the Harletz MIS 6 loess records reveals the occurrence of alternating loess intervals and paleosols, similar to those from the last climatic cycle in Europe. These paleosols are correlated with interstadials described in various Mediterranean records and correspond to warmer and moister events, which may also correlate with other Northern Hemisphere interstadials. In Northwestern Europe, MIS 6 tundra gleys have also been described in loess sequences. Locht et al. (2016) report three tundra gleys and one paleosol in Northern France. In Belgium, Juvigné et al. (1996) describe five tundra gleys in Kesselt, while Pirson et al. (2018) report two tundra gleys in Albert Canal and Haesaerts et al. (2016) mention one and two tundra gleys in Harmignies and Remicourt, respectively. Finally, in the lower Rhine valley in Germany, Schirmer (2010), defining a synthetic Rhine pedostratigraphical record, refers to eight tundra gleys over two thin MIS 6 paleosols (a basal humus zone and a calcareous cambisol). However, the precise dating of all these tundra gleys among MIS 6 remains unclear without any particular investigation and therefore prevent correlations as defined for the last climate cycle.





The Greenland ice-cores are the key paleoclimatic references used to interpret the paleosol-loess unit alternations identified during the last climate cycle in the European loess sequences at about 50°N. They all show a very precise imprint of the abrupt changes over the last 130 ka. Among them, the NGRIP $\delta^{18}O$ record has an extremely high resolution although layer counting errors accumulate down the core (Boers et al., 2017), yielding suitable dates for the onset and end of the abrupt
warmings as observed in various parameters (Fischer et al., 2015; Schupbach et al., 2018), and therefore allows precise correlations with the terrestrial sequences (Moine et al., 2017; Rousseau et al., 2002; 2007; 2017a; 2017b).
Although initially described in the GRIP record (Dansgaard et al., 1993), no high-resolution record of the penultimate climate cycle is available from the Greenland ice-sheet. Moreover, the Antarctic ice-core records are not directly usable in a first step to compare them with the European loess sequences (see below). A potential solution therefore is given by
considering the long high-resolution records from Chinese speleothems. Indeed, the Hulu cave $\delta^{18}O$ record has been demonstrated to perfectly correlate with the NGRIP record of the last climate cycle ( Wang et al., 2001; Cheng et al., 2006) through precise [230]Th ages. Moreover, this precise and detailed climate record has been extended back in time until 224 ka b2k through the addition of the measured $\delta^{18}O$ performed on the Sanbao cave speleothem SB11, which has resolution similarly high to that of the Hulu cave, but for the penultimate climate cycle. This is the reason why we use this particular
record as a reference for our comparison

### 4.1 Sanbao11 (Fig. 3, Supp. Fig. 1)

The Sanbao cave is located at 110° 26' E - 31° 40' N, 1,900 m a. s. l., in central China, at the edge of the Chinese Loess Plateau. It is strongly influenced by the East Asian monsoon variations and correlates with the summer insolation gradient between 65°N and 15°N. Moreover, similarly to the last climate cycle record measured from the Hulu Cave, the $d^{18}0$ record
of Sanbao speleothem B11 indicates abrupt changes at millennial scale during MIS 7 and 6, which have been interpreted as interstadials by Wang et al (2008). Considering the age boundary between MIS7 and 6 (Lisiecki and Raymo, 2005) at 191 ka, the Sanbao speleothem preserved 5 interstadials, named B24 to B20, in MIS7 (although missing the lower 19 ka –limit at -243 ka) and 13, named B19 to B1, in MIS6. The $\delta^{18}O$ minima in MIS 6 are dated at -189 ka for B19, -181 ka for B18, -175.5 ka for B17, -172 ka for B16, -168 ka for B15, -166 ka for B14, -162 ka for B13, -160 ka for B12, -157 ka for B11, -151 ka
for B10, -149 ka for B9, -147.5 ka for B8, -145 ka for B7, -136 ka for B2 and -134 ka for B1 (Fig. 3); note that B3 to B6 are better recorded in the Hulu Cave MSP speleothem (Cheng et al., 2006). According to the prevailing interpretation, the lower the $d^{18}0$ values from the speleothem record, the more intense was the East Asian summer monsoon intensity. Interestingly, the minimum $\delta^{18}O$ values for B17, B16, and B15 in MIS6 are much lower than the other minimum values identified in the past 224 ka record, including those characterizing the 2 past interglacials (MIS 7 and 5e) and the Holocene one. Hence, the
time interval between -178 ka and -165 ka must have corresponded to extremely high precipitation rates over the East Asian region. This observation is in accordance with the high precipitation interval derived from the speleothems from Soreq and Argentarola caves in the Mediterranean region during sapropel 6. Interestingly, the Sanbao Cave record yields 5 interstadials between -190 ka and -167 ka, and hence the same number of interstadials as the number of paleosols observed in Harletz below the tephra layer. Moreover, the B21, B16 and B8 $\delta^{18}O$ minima correspond to peaks of decreasing magnitude in
summer insolation at 15°N. However, although summer insolation at both 15°N (tropics) and 65°N (arctic circle) show maximum values corresponding to the Sanbao $d^{18}0$ main minima, the insolation gradient (15°-65°), which induces a potentially strong meridional heat transport in the atmosphere and a possible excess in precipitation over the continent, indicates the weakest maximum for the time interval B17-B15 (about -177 ka – -166 ka) (Fig. 3).

### 4.2 Greenland GL$_t$_syn (Fig. 1, 3, Tab. 1).





Using the argument of the bipolar seesaw interpretation of the anti-phased climate variation prevailing between Antarctica and Greenland, Barker et al. (2011) have developed a synthetic d$^{18}$0 record of Greenland for the last 800 ka, based on EPICA results and time scale, named GL$_t$_syn. This synthetic record replicates nicely the millennial scale variability observed during the last climate cycle in the NGRIP ice-core and therefore makes the authors confident that it reproduces reliably the last 800 ka of Greenland d$^{18}$0 abrupt variability. Two time scales have been proposed, one based on EPICA DC3 and another

based on the Sanbao speleothem, with some discrepancies. Because of the more precise dates obtained from the Chinese speleothems, we will use the GL$_t$_syn with the Sanbao time scale for our comparisons. Plotted versus the Sanbao d$^{18}$0 record, the GL$_t$_syn record indicates intervals showing peak maximum values aligned with most of the Chinese speleothem interstadials. Among these peak values, Barker et al (2011) predicted the occurrence of eleven DO-like events, which should correspond to as many interstadials (Fig. 3, Tab. 1). Five of them can be clearly identified between -190 ka and -166 ka,

which align with the Sanbao ones, supporting therefore the global value of Harletz paleosols interpreted as interstadials.

### 4.3 North Atlantic (Fig. 1, 3, Tab. 1, Supp. Fig. 1)

South of Greenland, the North Atlantic Ocean is the classical region where to observe the Greenland warmings/interstadials as this was demonstrated for the last climate cycle (Bond et al., 1992; McManus et al., 1999; Henry et al., 2016; Hodell and Channell, 2016).

For the penultimate cycle, several records also indicate warming events interpreted by the respective authors as interstadials. The warming events observed in the SST, deduced from the alkenone studies from the Iberian margin, are of similar magnitude as those observed in the Alboran Sea (ODP977/MD12443 sea above) (Fig. 3). Furthermore, the Iberian margin core MD01-2444 shows temperate tree pollen percentages, which vary in line with the planktonic δ$^{18}$O measured in the same core (Margari et al., 2014). Such high percentages of arboreal pollen during the base of MIS6 (-185ka to -160ka) are

interpreted in terms of reduced summer aridity and increased pluvial conditions. Back to the sea surface temperature, which has been interpreted as mimicking the Greenland δ$^{18}$O variation (Shackleton et al., 2000), plotting ODP977 alkenone SST reconstructions of MIS7 and 6 against GL$_t$_syn, one can notice numerous discrepancies appear in the number of interstadials and for those identical in their occurrences, which seem to be due to the age model used for ODP 977. Further north, in core U1308 (49°53'N – 24°14W, 3871 m nsl) Obrochta et al. (2014) report important variations in *Neogloboquadrina*

*pachyderma sinistra*, an indicator of cold surface water, during MIS6. The minimum counts are much stronger at the base of the MIS6 record, between -190 ka and -160 ka, than in its upper part. Although dating these warming events can be proposed from the published material, at about -190ka, about -175.5ka, -167.5ka, about -160 ka, about -166.5 ka, about -143 ka and about -137 ka, these dates remain preliminary as the time resolution is not high enough compared to GL$_t$_syn or ODP977/MD01-2443 marine cores (Fig. 3). M23414-9 gravity core (53.537°N, 20.288°W; water depth 2199 m), selected to

study variations in the North Atlantic drift, indicates the highest values in *N pachyderma s.* percentages between -165 ka and -130ka. However, minima correspond to warmings in both summer and winter sea surface temperature estimates (Kandiano et al., 2004). Eleven of such events can be dated at about -192 ka, -187.5 ka, -180.5 ka, -175.5 ka, -172 ka, -163.5 ka, -158.5 ka , -141 ka, -140 ka, -137 ka and -133.5 ka BP, but show very few overlapping with the other marine cores previously mentioned, probably because of a lower resolution. Still, the five lowest occurred before -165 ka. A similar pattern is noticed

further north, in ODP983 (60.4°N – 23.6°W, 1,984 m nsl) with stronger minimum counts of N. *pachyderma* s. between 190 ka and 160 ka than between -160 ka – -130 ka. Using the "GICC05/NALPS/China" timescale (Barker et al., 2015), warming events are observed at about -189.9 ka, -178.7 ka, -173.2 ka, -169.5 ka, -168.2 ka, -163 ka, -160.7 ka, -151 ka, -149.1 ka, -147.4 ka, -140.4 and -134.5 ka (Fig. 3). These dates vary slightly when using the other proposed time scale based on EDC3 age model. Closer to Greenland and south of Iceland, ODP 984, drilled at (61.25°N and 24.04°W, 1648 m), yielded a record

of the past 220 ka (Mokeddem and McManus, 2016). It includes a complete MIS6, showing millennial-scale variability that is best expressed during the interval -170 ka to -130 ka. The foraminifer composition indicates the alternation between



episodes of northward polar-front retreats (low values of *N pachyderma s.*), synchronous with warm and salty water inflows (high values of *Turborotalia quinqueloba*), and episodes of southward polar-front advance (high values of *N pachyderma s*), with fresh and cold water inflows (low *T quinqueloba*) and high IRD content. The cold inflow episodes are interpreted as

stadials while the warm ones are considered as interstadials equivalent to those observed during the last climate cycle. They occurred at about -165 to -163 ka, about -156 to -152 ka, about -148 to -146 ka, about -139 to -137 ka, and about -135 to -132 ka, and are named IS6-5, IS6-4, IS6-3, IS6-2, IS6-1 respectively (Fig. 3). Interestingly, these interstadials are the nearest geographically described to Greenland during MIS6, but considering the resolution, more events could be considered when looking at the percentage of *N pachyderma* left (see figure 3 and table 1).

When comparing the MIS6 records described previously, e. g. the reconstructed Greenland d$^{18}$0, the Sanbao speleothem d$^{18}$0, as well as the foraminifer counts and d$^{18}$0 measurements from Sites 984 and 983, one can notice that the interstadials observed in the North Atlantic records can have an equivalent in both the reconstructed Greenland synthetic and Sanbao d$^{18}$0 interstadials (Fig. 3). Plotting every record used in the present study on its individual time scale, a global synchronism is far from being evident, except for the early MIS6. When reconstructing the Greenland d$^{18}$0 variations, Barker et al. (2011)

predicted DO event occurrences for the last climate cycle, which fit with the DO events described from the Greenland ice-cores. They expanded the occurrence prediction to the previous climate cycles covering the past 800ka. Plotted versus the Sanbao interstadials, although these predicted DO-like events fit for the early part of MIS6, between -192 and -166 ka, fewer abrupt changes are assigned D-O like labels than the Sanbao record yields interstadials for the -166 - -130 ka interval, especially in the upper (more recent) MIS6. This is demonstrated when comparing the reconstructed Greenland d$^{18}$0, the

variations in *N. pachyderma* left percentages in the North Atlantic ODP984, the closest to Greenland, but also U1308 records. Reporting all the interstadials in table 1 supports the previous interpretation that these interstadials are hemispheric and can be interpreted as DO-like events, without any doubt at least for the early MIS6. However, the question remains why these marine records did not show all of the interstadials described in the other domains, contrary to what has been observed during the last climatic cycle. Is this related to the time resolution of the studied cores, which would not be high enough?

This is an on-going problem that this paper cannot address, as it requires further investigations and additional data with higher resolution to be obtained.

### 4.4 Chinese Loess records (Supp. Fig. 1)

One potential record of all these events has been proposed from sequences from the Chinese Loess Plateau, located north of the high-resolution speleothems described previously.

For the last glacial cycle, the study of the grain size variations from temporally highly resolved loess sequences indicates changes in the size of the deposited particles which are aligned, using a time scale based on luminescence dates, with the DO events (Sun et al., 2012). As the deposited material originates from Northern Chinese deserts, during interstadials, when the wind velocity reduces due to stronger East Asian monsoon blowing from the South, finer material is deposited. On the contrary, during stadials, the wind velocity is stronger and the East Asian summer monsoon weakens, allowing coarser

material to deposit. Following this idea, Yang and Ding (2014) have proposed a stack of several loess sequences covering the last two climate cycles (Supp. Fig. 1). They show that for the last 130 ka, the median grain size variations match the former results by Sun et al. (2012) and correlate with the Hulu Cave speleothem d$^{18}$0 record (Cheng et al., 2006; Wang et al., 2008). Expanding to the penultimate cycle, Yang and Ding (2014) indicate that the median grain size variations during MIS 7 and 6 also mimic the Sanbao cave d$^{18}$0 variations, within the uncertainties related to the dating of these sequences.

Interestingly, the authors refer to interstadial/paleosols, which do not correspond to the identified individual interstadials but rather to a group of them (L2-4 groups B17, B16 and B15 while L2-2 gathers B11 to B5). This observation is similar to what was described for the last climate cycle record by Sun et al (2012), where not every interstadial is individually resolved in





the stratigraphy, contrary to the various paleosols in the European loess sequences at about 50°N, which correspond one-to-one with the Greenland interstadials.

**5 Concluding comments**

The identification of interstadials/paleosols in the Harletz MIS6 record, which were not known previously in the loess records of the nearby sequences, led us to evaluate the significance of these warm and moist episodes. Such events appear to have equivalents in the northern Mediterranean region and are expressed through different parameters. Interestingly, some of these events correspond to very humid episodes as recorded in speleothems, corresponding to the deposition of Sapropel 6 in

the Mediterranean Sea, associated with a northward shift in the position of the summer ITCZ. North Atlantic cores also yield evidence of these interstadials, although the available records are not complete except at the vicinity of the Gibraltar Strait. In the Iberian Sea, similar events in the SST reconstructions based on alkenones as in the Alboran Sea can be observed. These North Atlantic MIS6 interstadials are also associated with lower benthic $d^{13}C$ values, inducing continental uptake of the $^{12}C$ by the development of vegetation increase, in agreement with synchronous arboreal peaks in the pollen records, and

synchronous soil development.

At a broader geographic scale, these warm events are also well recorded synchronously in Chinese speleothems and the Greenland synthetic $\delta^{18}O$ variations of MIS6. Therefore, contrary to the initial interpretation as regional events, the ISm and ISp interstadials mostly expressed in Harletz provide evidence of events at least at the northern hemisphere. In Greenland, they correspond to strong warmings, in the North Atlantic to abrupt SST increases, partly development of incipient

weathered horizons and paleosols in European loess, arboreal vegetation in European, mostly Mediterranean, pollen records, minimum values in the $\delta^{18}O$ from Mediterranean and Chinese speleothems, and maxima in Chinese stack median grain-size. There is, therefore, a similarity between these interstadials described from MIS6 records in the Atlantic, Asian and European regions including the Mediterranean area, and those described for the last climate cycle in the same regions.

In summary, we argue that there are similarities between the well-established, globally synchronous interstadials of the last

glacial cycles, and corresponding episodes of the penultimate glacial cycle. This is true for the local character of these episodes in the different kinds of records, ranging from marine and lake sediments to ice cores and loess sequences, but there is rising evidence that it is also true for the global synchronicity of the interstadials identified in the different records. Clearly, the available data for the penultimate cycle are substantially sparser, have much coarser resolution, and are overall less reliable. Nevertheless, taken together, the available empirical evidence suggests that the MIS6 interstadials described in

the various environments, and correlated in a similar way as for the last 130 ka, i.e., the various paleosols described at the base of MIS6 in Harletz and corresponding to the interval 192-166 ka, can confidently be interpreted as DO-like events of the penultimate climate cycle. With improved resolution of records of the penultimate glacial, and particular more detailed dating, a thorough statistical analysis of the synchronicity of interstadials in the different records may become possible in the near future. Table 1 shows that uncertainties in the different employed age models have probably resulted in many of the

interstadials from one record not lining up with those from another record. Therefore, additional methodological developments will also be required for that purpose, given that the different records have potentially very different age models, time scales, dating and other types of uncertainties.

At least the last two climate cycles, and probably further previous ones, are nevertheless likely to have experienced millennial scale variability around the globe, which cannot not be directly explained by any astronomical forcing. Such

climate variability may have been induced by forcings superimposed to the classical orbital parameters inducing the climate cycles, and is most likely caused by self-sustained oscillations induced by interactions between ice cover and ocean circulation changes.

**Author contribution**



DDR designed the study and prepared the manuscript with contributions from all co-authors. PA and FL conducted the
fieldwork. PA performed the pedostratigraphy, FL the environmental magnetism and MD the spectroscopy analysis.

**Acknowledgments**

The comments from Peter Clark on a preliminary version of this manuscript were very useful and welcomed. Thanks to
Kathy Roucoux for providing the Ionnina I-284 data, Steve Barker for providing the Greenland $GL_t$_syn and ODP 983 *N
pachyderma s* data. This study benefited from research funds granted by the ANR to DDR (ACTES project ANR-08-BLAN-
0227 - CSD 6, CNRS-INSU LEFE program to DDR), the CNRS-INSU SYSTER program to FL (2012-31124A) and the
PHC Rila program to FL and DJ (34286QB). This work is performed under the TiPES project funded by the European
Union's Horizon 2020 research and innovation program under grant agreement # 820970. This is LDEO contribution zzz,
IPGP contribution xxx, and TiPES contribution yyyy

**Datasets used:**

•   Obrochta, Stephen P; Crowley, Thomas J; Channell, James E T; Hodell, David A; Baker, Paul A; Seki, Arisa;
Yokoyama, Yusuke (2014): Coarse grain counts in MIS6 of IODP Site 303-U1308. PANGAEA,
https://doi.org/10.1594/PANGAEA.834636,
•   Bar-Matthews, M., et al., 2003, Soreq and Peqiin Caves, Israel Speleothem Stable Isotope Data,IGBP
PAGES/World Data Center for Paleoclimatology , Data Contribution Series #2003-061. NOAA/NGDC
Paleoclimatology Program, Boulder CO, USA.
•   Wang, Y., et al. 2009. Sanbao Cave, China 224 KYr Stalagmite ™18O Data. IGBP PAGES/World Data Center for
Paleoclimatology, Data Contribution Series # 2009-138. NOAA/NCDC Paleoclimatology Program, Boulder CO,
USA.
•   Sadori, Laura; Koutsodendris, Andreas; Panagiotopoulos, Konstantinos; Masi, Alessia; Bertini, Adele;
Combourieu-Nebout, Nathalie; Francke, Alexander; Kouli, Katerina; Kousis, Ilias; Joannin, S√©bastien; Mercuri,
Anna Maria; Peyron, Odile; Torri, Paola; Wagner, Bernd; Zanchetta, Giovanni; Sinopoli, Gaia; Donders, Timme H
(2018): Pollen data of the last 500 ka BP at Lake Ohrid (south-eastern Europe). PANGAEA,
https://doi.org/10.1594/PANGAEA.892362
•   Martrat, Belén; Grimalt, Joan O; López-Martinez, Constancia; Cacho, Isabel; Sierro, Francisco Javier; Flores, José-
Abel; Zahn, Rainer; Canals, Miquel; Curtis, Jason H; Hodell, David A (2004): Sea surface temperatures, alkenones
and sedimentation rate from ODP Hole 161-977A. PANGAEA, https://doi.org/10.1594/PANGAEA.787811,
Supplement to: Martrat, B et al. (2004): Abrupt Temperature Changes in the Western Mediterranean over the Past
250,000 Years. Science, 306(5702), 1762-1765, https://doi.org/10.1126/science.1101706
•   Martrat, Belén; Grimalt, Joan O; Shackleton, Nicholas J; de Abreu, Lucia; Hutterli, Manuel A; Stocker, Thomas F
(2007):   Sea   surface   temperature   estimation   for   the   Iberian   Margin.   PANGAEA,
https://doi.org/10.1594/PANGAEA.771894, Supplement to: Martrat, B et al. (2007): Four climate cycles of
recurring deep and surface water destabilizations on the Iberian Margin. Science, 317(5837), 502-507,
https://doi.org/10.1126/science.1139994
•   http://www1.ncdc.noaa.gov/pub/data/paleo/contributions_by_author/mokeddem2016/mokeddem2016-odp984.txt
•   Kandiano, Evgenia S (2009): (Table A.01) Planktic foraminiferal census data of sediment core GIK23414-9.
PANGAEA, https://doi.org/10.1594/PANGAEA.713761
•   Antoine, P (2019): High resolution (5 cm) continuous record of grain size and magnetic susceptibility of the 20 m
thick loess palaeosol sequence of Harletz, Bulgaria. https://doi.pangaea.de/10.1594/PANGAEA.905216



- Antoine, P (2019): High resolution continuous record of color (a*) of the 20 m thick loess palaeosol sequence of
Harletz, Bulgaria. https://doi.pangaea/10.1594/PANGAEA.905214

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



**Figure captions**

Fig. 1. Harletz loess sequence. A, Location of the western sites discussed in the text in the MIS6 maximum ice sheet extent by F. Colleoni from (Dendy et al., 2017), the black box refers to B. B, Location of Harletz loess sequence (red dot) in Bulgaria and nearby loess sequences (black dots) cited in the text (from (Antoine et al., 2019) modified; 1: eolian sands, 2: sandy loess, 3 loess>5m, 4 loess<5m). C, Pedotratigraphy and variation of some parameters (grain-size, magnetism, color reflectance, from (Antoine et al., 2019), modified). Interstadials deduced from the measured parameters (ISm) and corresponding to incipient weathered horizons and from paleosol observation (ISp) (this study).

Fig. 2. Western (ODP 977 SST –alkenone, (Martrat et al., 2004)) to eastern (Soreq speleothem $\delta^{18}$O , (Bar-Matthews et al., 2003)) Mediterranean records of MIS6, including pollen record from Ionnina I-284 (AP, AP-Pine+Birch+Juniper, (Roucoux et al., 2011)) compared with summer insolation with indication of the time occurrence of Sapropel 6 (Bard et al., 2002). Labels of the interstadials identified in the Alboran Sea. Complementary labels assigned warm events of ODP 977 (AI-5'a,b, AI-9' a, b), Ionnina (I1-12) and Soreq (S1-9) Mediterranean MIS6 records used in Table 1. MIS6 ice sheet extent by F. 630    Colleoni in Dendy et al. (2017).

Fig. 3. Comparison of northern Hemisphere records of MIS6. Greenland synthetic $\delta^{18}$O (Barker et al., 2011), North Atlantic SST (N. *pachy* s) (cores ODP984 –(Mokeddem and McManus, 2016)), ODP983 –(Barker et al., 2015), U1308 –(Obrochta et al., 2014)), Iberian margin and Alboran Sea SST (alkenone MD01-2443, ODP 977-(Martrat et al., 2004; 2007)), and Chinese speleothem $\delta^{18}$O (Sanbao11 –(Wang et al., 2008)). Labels of the interstadials identified in the marine and speleothem 635    records. Green dots denote predicted D-O event occurrence from Barker et al. (2011). Complementary labels assigned warm events of ODP 984 (IS6-3a,b), U1308 (1-8), ODP977/MD01-2443 (AI-5'a,b; AI-9'a,b) North Atlantic and Mediterranean MIS6 records used in Table 1. MIS6 ice sheet extent by F. Colleoni in Dendy et al. (2017).

Table 1. Synchronism of the various interstadials discussed in this study. Indication of the predicted D-O event occurrence by Barker et al. (2011), of the various assigned interstadials from Sanbao11 and Soreq speleothem $\delta^{18}$O , ODP 984, 983 and 640    U1308 percentage in *N pachyderma* left, ODP977 and MD01-2443 SST and Ionnina I-284 arboreal pollen variations. The marine isotope stage (MIS) 6 is decomposed into an early part corresponding to the occurrence of the Harletz paleosols (192-170 ka), and a later part (170-130 ka) with incipient weathered horizons





Fig. "1"

Fig. 2

Fig. 3





Climate of the Past Discussions — Open Access — EGU

Tab. 1

Predicted D-O event occurrence

| Age kyr (EDC3) | SpeloAge (kyr) | DO pick | DO pick variable threshold | Sanbao11 | ODP984 | ODP983 | U1308 | ODP977/ MD01-2443 | Ionnnina | Soreq | Harletz | MIS6 |
|---|---|---|---|---|---|---|---|---|---|---|---|---|
|  |  |  |  | B01 | IS6-1 |  |  | AI-1' | I1 | S1 |  |  |
|  |  |  |  | BO2 | IS6-2 | x | U1308-1 | AI-2' | I2 | S2 |  |  |
|  |  |  |  | B07 | IS6-3b |  | U1308-2 |  | I3 | S3 |  |  |
|  |  |  |  |  | IS6-3 |  |  |  |  |  |  |  |
| 146,70 | 147,37 | 1 | 1 | B08 | IS6-a |  |  |  |  |  |  |  |
| 148,30 | 149,08 | 0 | 1 | B09 |  | x | U1308-3 | AI-3' |  | S4 |  |  |
| 150,06 | 150,97 | 1 | 1 | B10 | IS6-b |  | U1308-4b | AI-4' |  |  |  | MIS6 P2: 130-170 ka |
|  |  |  |  |  | IS6-4 |  |  |  |  |  |  |  |
| 159,12 | 160,69 | 1 | 1 | B11 |  |  | U1308-4 | AI-5'-b | I4 | S5 |  |  |
|  |  |  |  | B12 |  | x | U1308-5 | AI-5'-a | I5 |  |  |  |
| 162,16 | 164,03 | 1 | 1 | B13 | IS6-5 | x | U1308-6b | AI-6' | I6 | S6 |  |  |
|  |  |  |  | B14 |  |  | U1308-6 | AI-7' | I7 | S7 |  |  |
| 168,16 | 169,48 | 1 | 1 | B15 |  | x |  |  |  |  |  |  |
|  |  | 5 | 6 | 15 | 8 | 5 | 8 | 8 | 7 | 7 | 5 | # DO P2 |
| 172,00 | 173,18 | 1 | 1 | B16 |  |  | U1308-7 | AI-8' | I8 | S8 |  |  |
| 174,98 | 176,29 | 1 | 1 | B17 |  |  | U1308-8 | AI-9'-b | I9 |  |  | MIS6 P1: 170-192ka |
| 177,14 | 177,96 | 1 | 1 |  |  | x |  |  |  |  |  |  |
| 178,30 | 178,65 |  |  | [B18] |  | x |  | AI-9'-a | I10 | S9 |  |  |
|  |  |  |  | B18 |  |  | U1308-9 | AI-10' | I11 |  |  |  |
| 187,84 | 189,98 | 1 | 1 | B19 |  | x | x | x | I12 |  |  |  |
| 192,16 | 192,68 | 1 | 1 | B20 |  | x |  |  |  | S2 | 5 | # DO P1 |
|  |  | 5 | 5 | 5 | 4 | 4 | 4 | 5 | 5 | 2 | 5 |  |
|  |  | 10 | 11 | 20 | 8 | 9 | 12 | 13 | 12 | 9 | 10 | # DO MIS6 |

| Max/Min | Nb | frequence (ka) |  |
|---|---|---|---|
| max: | 15 | 2,67 | MIS6 P2: 130-170 ka |
| min: | 4 | 10,00 |  |
| max: | 5 | 4,40 | MIS6 P1: 170-192ka |
| min: | 2 | 11,00 |  |
| max: | 20 | 3,10 | # DO MIS6 |
| min: | 7 | 8,86 |  |