# Peer review of "DO-like events of the penultimate climate cycle: the loess point of view"

_Climate of the Past, 2019_

## Referee Comment (RC1) · Anonymous Referee #1 · 12 Oct 2019

General comments: The Dansgaard-Oeschger (DO) events in Greenland ice-cores for the last glaciation have been well studied on a global scale. Whether such millennial-scale climate variability occurred during the preceding glaciations, however, remains less clear due to the limited length of the ice-cores and the low-resolution and dating uncertainty of terrestrial and marine records. In this study, the authors investigated the alternation of loess deposits and paleosol units of the Harletz loess sequence in Bulgaria formed during the penultimate glaciation and compared this succession with other regional proxy records to examine the occurrence of DO-like events prior to the last glaciation. The data and inferences presented in this study are of great significance and would contribute to understanding spatial coherences of the millennial-scale climate variability in the context of different glaciations. I recommend acceptance of this

manuscript for publication in CP after minor revisions.

Specific comments: 1. Line 115Âň. I suggest that the subtitle "3. Comparison with closest Mediterranean records" would be changed to "3. Proxy records in the Mediterranean region". 2. Lines 130-132Âň. What makes this record remarkable? Position, topography, or something else? The readers would be interested in it. Give a brief explanation, please. 3. Line 195Âň. I suggest that the subtitle "4. Paleoclimatology: DO-like events during MIS6" would be changed to "4. DO-like events during MIS6 in different regions". 4. Lines 222-223. Delete "at the edge of the Chinese Loess Plateau". Between the Loess Plateau and the Sanbao cave there lie two west-east extended mountains. 5. Lines 330-367. As the conclusion, this part appears too long. I suggest that the sentences on lines 359-367 would be removed. 6. Figs. 2 and 3. In these two figures, no proxy data of the Harletz loess sequence are shown. I suggest that one typical proxy of the Harletz loess sequence could be added into Figs. 2 and 3 in order to help the readers to compare the Harletz loess sequence with other records.

Related aspects: 1. Does the paper address relevant scientific questions within the scope of CP? Yes. 2. Does the paper present novel concepts, ideas, tools, or data? Yes. 3. Are substantial conclusions reached? Yes. 4. Are the scientific methods and assumptions valid and clearly outlined? Yes. 5. Are the results sufficient to support the interpretations and conclusions? Yes. 6. Is the description of experiments and calculations sufficiently complete and precise to allow their reproduction by fellow scientists (traceability of results)? Yes. 7. Do the authors give proper credit to related work and clearly indicate their own new/original contribution? Yes. 8. Does the title clearly reflect the contents of the paper? Yes. 9. Does the abstract provide a concise and complete summary? Yes. 10. Is the overall presentation well structured and clear? Yes. 11. Is the language fluent and precise? Yes. 12. Are mathematical formulae, symbols, abbreviations, and units correctly defined and used? Yes. 13. Should any parts of the paper (text, formulae, figures, tables) be clarified, reduced, combined, or eliminated? No. 14. Are the number and quality of references appropriate? Yes. 15. Is the amount

and quality of supplementary material appropriate? Yes.

---

## Author Comment (AC1) · 29 Oct 2019

We would like to thank reviewer 1 for his comments and are replying to some of them in the following lines.

Concerning the changes in the subtitles 3 and 4 we agree upon and will change them in the revised version.

Concerning comment 2, we are referring to the title of the paper we published previously about this record (Antoine et al. 2019). In fact the sentence is probably not clear enough, but what is remarkable in this record is first of all that the deposit corresponding to the penultimate glacial is highly developed and moreover it is recording such high number of paleosols and pedogenic horizons which has not at all been described

either westwards in Serbia or eastwards along the Danube river. We are adding the brief explanation and a figure in the supplementary material as required.

The stratigraphy of the MIS6 record from the Harletz sequence has been added in a revised version of Figures 2 and 3 at the request of reviewer 1.

denis-didier Rousseau on behalf of the co-authors of the manuscript

---

## Referee Comment (RC2) · Anonymous Referee #2 · 3 Dec 2019

While the DO cycles were widely recognized in globally distributed archives, the nature of abrupt climate changes remains controversial before the last climate cycle. This paper presents a good review on DO-like events of the penultimate climate cycle. New investigations of European loess records from MIS 6 reveal the occurrence of alternating loess intervals and paleosols, similar to those from the last climatic cycle and to those in lake, marine and speleothem records. Based on statistical similarity between these millennial-scale oscillations, the authors argue that the abrupt cause and global imprints were persistent during at least the last two climate cycles. This topic is suitable for the scope of the CP, but current version can be improved if the following concerns can be fully incorporated in a revised version.

1ïïjŐTitle: "The loess point of view" can be removed from the title, since this paper

includes a broad review of DO-like events in different records.

2. Add a new map showing the locations of loess, speleothem, lake, and marine records mentioned in this paper, rather as separated maps in each figure.

3. Fig.1 and related text: As a review of European loess records, the authors emphasized the similarity of DO-like events during the last two climatic cycles. But this similarity can not be verified from a single loess record shown in Fig.1. It's very necessary to add additional loess records with clear DO events of the last climatic cycle in Fig.1 to confirm the similar expression of millennial events during last two glaciations. While several weak paleosol layers can be identified in the outcrop, some layers in early MIS 6 can be easily judged by abrupt proxy changes, but others in late MIS 6 is not evident. Please clarify how many DO-like events can be robustly confirmed from the loess proxies, which is the key for further comparison with other records.

4. Fig.2 and related text: A regional synthesis of high-resolution records from adjacent lakes and ODP sites can confirm the presence of DO-like events in the MIS 6. Two concerns need to be clarified: (1) why the Soreq d18O record is different from those of lake and marine records (e.g., the precession cycles and DO-like events); and (2) How to correlate the DO-like events of S1-9 and I1-12 to those in the Haletz loess sequence.

5. Fig.3 and related text: A global synthesis of abrupt events in the MIS 6 is presented in Fig.3. It seems to me that the magnitudes and timing of these abrupt events are quite different. I would suggest employing a unified strategy to synchronize and numbering these DO-like events, rather than just putting these records together. Then, the similarity and discrepancies among these records can be properly addressed, which permits a better understanding of the abrupt cause and global nature of these DO-like events.
* * *

---

## Author Response (AR1)

**Replies to Reviewers 1 and 2**

**Reviewer 1**

General comments: The Dansgaard-Oeschger (DO) events in Greenland ice-cores for the last glaciation have been well studied on a global scale. Whether such millennial-scale climate variability occurred during the preceding glaciations, however, remains less clear due to the limited length of the ice-cores and the low-resolution and dating uncertainty of terrestrial and marine records. In this study, the authors investigated the alternation of loess deposits and paleosol units of the Harletz loess sequence in Bulgaria formed during the penultimate glaciation and compared this succession with other regional proxy records to examine the occurrence of DO-like events prior to the last glaciation. The data and inferences presented in this study are of great significance and would contribute to understanding spatial coherences of the millennial-scale cli-mate variability in the context of different glaciations. I recommend acceptance of this manuscript for publication in CP after minor revisions.

We would like to thank reviewer 1 for his comments and suggestions.

Specific comments:

1. Line 115. I suggest that the subtitle "3. Comparison with closest Mediterranean records" would be changed to "3. Proxy records in the Mediterranean region".

Concerning the changes in the subtitles 3 we agree upon and will change them in the revised version.

2. Lines 130-132. What makes this record remarkable? Position, topography, or something else? The readers would be interested in it. Give a brief explanation, please.

Concerning comment 2, we are referring to the title of the paper we published previ-ously about this record (Antoine et al. 2019). In fact the sentence is probably not clear enough, but what is remarkable in this record is first of all that the deposit correspond-ing to the penultimate glacial is highly developed and moreover it is recording such high number of paleosols and pedogenic horizons which has not at all been described either westwards in Serbia or eastwards along the Danube river. We are adding the brief explanation.

3. Line 195. I suggest that the subtitle "4. Paleoclimatology:DO-like events during MIS6" would be changed to "4. DO-like events during MIS6 in different regions".

Concerning the changes in the subtitle 4 we agree upon and will change them in the revised version.

4. Lines 222-223. Delete "at the edge of the Chinese Loess Plateau". Between the Loess Plateau and the Sanbao cave there lie two west-east extended mountains.

Correct. We have changed "at the edge of" by "South of

5. Lines 330-367. As the conclusion, this part appears too long. I suggest that the sentences on lines 359-367 would be removed.

We have removed the sentences but added a summary of the total number of DO-like interstadials identified in Harletz record.

6. Figs. 2 and 3. In these two figures, no proxy data of the Harletz loess sequence are shown. I suggest that one typical proxy of the Harletz loess sequence could be added into Figs. 2 and 3in order to help the readers to compare the Harletz loess sequence with other records

The stratigraphy of the MIS6 record from the Harletz sequence, with the identified interstadials has been added in a revised version of Figures 2 and 3, which are now Figures 3 and 4.

**Reviewer 2**

Thanks to reviewer2 for pointing key issues in this paper. We have tried to reply his main comments as follows in blue.

While the DO cycles were widely recognized in globally distributed archives, the nature of abrupt climate changes remains controversial before the last climate cycle. This pa- per presents a good review on DO-like events of the penultimate climate cycle. New investigations of European loess records from MIS 6 reveal the occurrence of alternat- ing loess intervals and paleosols, similar to those from the last climatic cycle and to those in lake, marine and speleothem records. Based on statistical similarity between these millennial-scale oscillations, the authors argue that the abrupt cause and global imprints were persistent during at least the last two climate cycles. This topic is suitable for the scope of the CP, but current version can be improved if the following concerns can be fully incorporated in a revised version.

1. Title: "The loess point of view" can be removed from the title, since this paper includes a broad review of DO-like events in different records.

Well Reviewer2's point is correct and could be adopted, however if we refer to our reply about point 5, this proposal remains questionable because we raise the issue based on our investigation on loess deposits.

2. Add a new map showing the locations of loess, speleothem, lake, and marine records mentioned in this paper, rather as separated maps in each figure.

This is already the case on Fig. 1 A and B but we will improve Fig. 1A by adding the name of the records to their precise location indicated by the red spots. However, they are all on a Google Maps plotted in supplementary figure 1.

3. Fig.1 and related text: As a review of European loess records, the authors emphasized the similarity of DO-like events during the last two climatic cycles. But this similarity can not be verified from a single loess record shown in Fig.1. It's very necessary to add additional loess records with clear DO events of the last climatic cycle in Fig.1 to confirm the similar expression of millennial

We agree with Reviewer2 that relying on one single record would be highly questionable. However, in the introduction of the study we recalled the correlations between paleosols, tundra gleys and incipient units we have made in European loess sequences at 50°N with the DO events determined in Greenland ice cores, based on precise dating of these pedogenic units. The similarity of the paleosol-loess succession at the base of Harletz high resolution MIS6 record with the LCC records in European loess sequences, allows us to infer that the warm events be considered as interstadials, therefore DO-like events. Furthermore, the pedogenic processes as expressed by the upper intervals inferred by the magnetism, grain-size and spectrocolor measurements lead us to include them among the identified interstadials. As the neighbor records in the Danube area, Fig. 1B, do show only one event expressed by a paleosol (former supplement Fig.2) we have decided to expand further our investigation into steps, first within the Mediterranean Basin, which shows a complex paleoclimate history, and secondly to more global records, including N European loess records.

While several weak paleosol layers can be identified in the outcrop, some layers in early MIS 6 can be easily judged by abrupt proxy changes, but others in late MIS 6 is not evident.

We agree with reviewer2 that this could be questionable, but magnetism measurements show a clear pedogenic effect as indicated by a normalized isothermal remanent magnetization proxy and bulk coercitivity of remanence occurring simultaneously to the variations, even slight in other proxy records like grains-size and spectrocolorimetry.

Please clarify how many DO-like events can be robustly confirmed from the loess proxies, which is the key for further comparison with other records.

As requested by reviewer1, we added in the figure 2 an extract of the MIS6 stratigraphy of Harletz (see Fig. 1C) showing the various identified DO-like events, interstadials, as ISp deduced from direct paleosol observations (4 ISp are identified) and as ISm deduced from reliable proxy measurements (6 ISm are identified)

4. Fig.2 and related text: A regional synthesis of high-resolution records from adjacent lakes and ODP sites can confirm the presence of DO-like events in the MIS 6. Two concerns need to be clarified: (1) why the Soreq $\delta^{18}$O record is different from those of lake and marine records (e.g., the precession cycles and DO-like events);

The representation of the Soreq $\delta^{18}$O record is presented inversely than the classical way with low values to the left characterizing increased rainfall amount above the cave situated in central Israel. Bar-Matthews et al 2003 describing this speleothem record and another one in northern Israel concluded that they recorded the proxy signal of global and regional Eastern Mediterranean climate over the last 250 kyrs. The age model of Soreq is based on $230^{Th}$-U dating while the marine, partly, and continental cores are orbital tuned.

As already stated by Martrat et al. (2007) for the last four climate cycles in the western Mediterranean and the Iberian margin, the observed variations in sea surface temperature, expressed by UK'37, show a nonlinear response to external triggers of climate that are obliquity and precession. However, some interstadials are synchronous to Mediterranean sapropels, which are a direct response to orbital forcings. Rohling et al. (2015) presented an updated review of the present and past Mediterranean climate and oceanography with a clear differentiation between the western and eastern basins providing explanations why Soreq $d^{18}$O differs from the other records.

and (2) How to correlate the DO-like events of S1-9 and I1-12 to those in the Harletz loess sequence.

We will replot Fig. 2 to show Soreq record the classical way to ease the reading with Ionnina and ODP977 and the evaluation of the potential synchronous events, but still keeping the original time scales. Doing so, this highlights the uncertainties in the different age models used in the different records. Moreover, as requested by Reviewer1 we added Harletz stratigraphy to a revised version of the figure to support our interpretation.

5. Fig.3 and related text: A global synthesis of abrupt events in the MIS 6 is presented in Fig.3. It seems to me that the magnitudes and timing of these abrupt events are quite different.

Yes, we agree with reviewer2's comment, but we wanted to present the raw evidences.

I would suggest employing a unified strategy to synchronize and numbering these DO-like events, rather than just putting these records together.

We agree with reviewer2's comment, but synchronization is the topic for an upcoming paper in preparation as well as deciphering/homogenizing the various numberings presently prevailing in several records.

Then, the similarity and discrepancies among these records can be properly addressed, which permits a better understanding of the abrupt cause and global nature of these DO-like events.

In this figure, we have pointed the DO-like events as determined by Barker et al when reconstructing the Greenland $\delta^{18}$O variation for the eight last climate cycle based on the bipolar seesaw mechanisms applied to the EPICA Dome C record. We also plotted the Chinese composite speleothem record for MIS6 in which interstadials have been proposed and numbered, just showing that there is still some significant work to perform before proposing a reliable frame and record. The marine records are there to show also the complexity of the record of interstadials in the Northern Atlantic Ocean during MIS6 while there was quite some homogeneity for the last climate cycle. Some numbering has been proposed for ODP984 and 977/MD01-2443 that we have reported.

Having this in hand, we prefer to refer our study to a loess point of view, see reviewer2's

comment#1, as we remain on just evidencing the reliability of the abrupt changes in our loess records although expecting further synchronization in the upcoming months.

**DO-like events of the penultimate climate cycle: the loess point of view**

[revised manuscript text omitted]

Denis-Didier Rousseau 29/10/y 22:23

Denis-Didier Rousseau 29/10/y 22:25

Denis-Didier Rousseau 29/10/y 22:21

Denis-Didier Rousseau 10/2/y 07:21

Denis-Didier Rousseau 10/2/y 07:21

Denis-Didier Rousseau 10/2/y 07:22

planktonic foraminifera *Globorigena bulloides* shows variations with abrupt changes mimicking – for the last climate cycle (MIS 5 to 2) – those described in the Greenland ice core records. During the penultimate cycle (MIS 7-6, -245 to -130 ka) the foraminifera $\delta^{18}O$ record also shows abrupt warmings similar to the DO interstadials, which have also been observed in the variations of the Uk'37 alkenone index, a proxy for the sea surface temperature. Nine interstadials are therefore identified, named Alboran or Iberian Margin interstadials, although the older one, AI-9', seems to be a composite one, in which 3 different events could be discriminated. The onsets of these penultimate cycle interstadials are dated at about: -186 ka (-186 ka, -182 ka, -179 ka), -176 ka, -170 ka, -165 ka, -159 ka, -152 ka, -150 ka, -141 ka and -133 ka, respectively. They were labeled AI-9', AI-8', AI-7', AI-6', AI-5', AI-4', AI-3', AI-2', and AI-1', respectively. Similarities in terms of the number of warming events appear with the long Mediterranean pollen records mentioned previously. Although no tephra is identified, 5 events can be once more identified prior to -165 ka, as also observed in Harletz (Fig. 3, Tab. 1). Martrat et al. (2007) stated the observed variations in sea surface temperature, expressed by UK'37 for the last four climate cycles in the western Mediterranean, show a nonlinear response to external triggers of climate that are obliquity and precession. However, some interstadials are synchronous to Mediterranean Sapropels, which are a direct response to orbital forcings. Rohling et al. (2015) presented an updated review of the present and past Mediterranean climate and oceanography with a clear differentiation between the western and eastern basins which may have impacted the adjacent terrestrial regions.

**3.4 Speleothem records (Fig. 3, Tab. 1)**

Two Mediterranean speleothems provide records for MIS6. The Argentarola Cave, located at (42°23'8.79"N 11°3'59.17"E) by the Tyrrhenian Sea, shows an interval of low $\delta^{18}O$ values between 180 ka and 165 ka with minima at 178 ka, 172 ka and 168 ka, the latter being in phase with high latitude insolation. The 180 ka-165 ka interval is interpreted as corresponding to the Mediterranean sapropel S6 (Bard et al., 2002) (Fig. 3). The recorded variations are correlated with total organic content variations recorded in the two Mediterranean cores MD84641 and KC19C, indicating rather warm and pluvial conditions during this interval over the western Mediterranean. Eastward, in Soreq cave in Israel, at 31.458 N 35.038 E and 400 m a.s.l, a speleothem record covering the last 180 ka was obtained. The measured $\delta^{18}O$ record of MIS6 shows variations at about -178 ka, -170 ka, -165 ka, -160 ka, -152 ka, -149 ka, -144 ka, -138 ka and -134 ka (Fig. 2). The peaks at -178 ka and -152 ka show very low values, interpreted as corresponding to intense pluvial intervals over Southern Israel, with the former correlated with the start of Sapropel S6, while no particular sapropel is associated with the latter (Ayalon et al., 2002; Bar-Matthews et al., 2003) although referred to a monsoon index maximum (Melières et al. 1997). These records indicate high precipitation rates at least during the -180 ka to -165 ka interval, which also corresponds to the most clearly expressed interstadials/paleosols in Harletz. Describing Soreq and another record in northern Israel, Bar-Matthews et al. (2003) concluded that these speleothems recorded the proxy signal of global and regional Eastern Mediterranean climate over the last 250 kyrs. As figure 3 shows, the age model of Soreq is based on $230^{Th}$-U dating and agrees with Bard et al. (2002) results, while the marine, partly, and continental cores are orbital tuned.

Although the interstadials identified in the Harletz MIS6 sequence have not been described in the closest loess sections from the Carpathian Basin and lower Danube region, the overview of the various and diversified MIS6 records presented above clearly indicates that these paleosols and incipient weathered horizons correspond to climatic events that are recognized all along the Northern Mediterranean, and therefore have more than a local/regional significance. This shows that the climate mechanism proposed to explain the paleosol-loess alternations of the last climate cycle (Antoine et al., 2009; Boers et al., 2017;2018; Rousseau et al., 2002; 2007; 2017a; 2017b), in northern European sequences at about 50°N, seem to have prevailed already during the penultimate climate cycle. This suggests that these alternations are part of a more global

dynamics. Therefore, in the next step of our study we will look for similar events in other records outside the Mediterranean region, at a more global scale.

**4 DO-like events during MIS6 in different regions (Fig. 4, Tab. 1)**

[revised manuscript text omitted]

planktonic foraminifera *Globorigena bulloides* shows variations with abrupt changes mimicking – for the last climate cycle (MIS 5 to 2) – those described in the Greenland ice core records. During the penultimate cycle (MIS 7-6, -245 to -130 ka) the foraminifera $\delta^{18}O$ record also shows abrupt warmings similar to the DO interstadials, which have also been observed in the variations of the Uk'37 alkenone index, a proxy for the sea surface temperature. Nine interstadials are therefore identified, named Alboran or Iberian Margin interstadials, although the older one, AI-9', seems to be a composite one, in which 3 different events could be discriminated. The onsets of these penultimate cycle interstadials are dated at about: -186 ka (-186 ka, -182 ka, -179 ka), -176 ka, -170 ka, -165 ka, -159 ka, -152 ka, -150 ka, -141 ka and -133 ka, respectively. They were labeled AI-9', AI-8', AI-7', AI-6', AI-5', AI-4', AI-3', AI-2', and AI-1', respectively. Similarities in terms of the number of warming events appear with the long Mediterranean pollen records mentioned previously. Although no tephra is identified, 5 events can be once more identified prior to -165 ka, as also observed in Harletz (Fig. 3, Tab. 1). Martrat et al. (2007) stated the observed variations in sea surface temperature, expressed by UK'37 for the last four climate cycles in the western Mediterranean, show a nonlinear response to external triggers of climate that are obliquity and precession. However, some interstadials are synchronous to Mediterranean Sapropels, which are a direct response to orbital forcings. Rohling et al. (2015) presented an updated review of the present and past Mediterranean climate and oceanography with a clear differentiation between the western and eastern basins which may have impacted the adjacent terrestrial regions.

**3.4 Speleothem records (Fig. 3, Tab. 1)**

Two Mediterranean speleothems provide records for MIS6. The Argentarola Cave, located at (42°23'8.79"N 11°3'59.17"E) by the Tyrrhenian Sea, shows an interval of low $\delta^{18}O$ values between 180 ka and 165 ka with minima at 178 ka, 172 ka and 168 ka, the latter being in phase with high latitude insolation. The 180 ka-165 ka interval is interpreted as corresponding to the Mediterranean sapropel S6 (Bard et al., 2002) (Fig. 3). The recorded variations are correlated with total organic content variations recorded in the two Mediterranean cores MD84641 and KC19C, indicating rather warm and pluvial conditions during this interval over the western Mediterranean. Eastward, in Soreq cave in Israel, at 31.458 N 35.038 E and 400 m a.s.l, a speleothem record covering the last 180 ka was obtained. The measured $\delta^{18}O$ record of MIS6 shows variations at about -178 ka, -170 ka, -165 ka, -160 ka, -152 ka, -149 ka, -144 ka, -138 ka and -134 ka (Fig. 2). The peaks at -178 ka and -152 ka show very low values, interpreted as corresponding to intense pluvial intervals over Southern Israel, with the former correlated with the start of Sapropel S6, while no particular sapropel is associated with the latter (Ayalon et al., 2002; Bar-Matthews et al., 2003) although referred to a monsoon index maximum (Melières et al. 1997). These records indicate high precipitation rates at least during the -180 ka to -165 ka interval, which also corresponds to the most clearly expressed interstadials/paleosols in Harletz. Describing Soreq and another record in northern Israel, Bar-Matthews et al. (2003) concluded that these speleothems recorded the proxy signal of global and regional Eastern Mediterranean climate over the last 250 kyrs. As figure 3 shows, the age model of Soreq is based on $230^{Th}$-U dating and agrees with Bard et al. (2002) results, while the marine, partly, and continental cores are orbital tuned.

[revised manuscript text omitted]

New Fig. "1"

New Fig. 2

[Figure]

New suppl. Fig. 2

[Figure]

Ice sheet extent by F. Colleoni from Dendy et al., 2017

Martrat et al., 2004

Bar-Matthews et al., 2003

**Age** (kyr) **ODP977** Uk'37 SST (°C)

**Soreq** δ¹⁸O (‰)

**Summer insolation 65°N** (W/m2)

**Age** (kyr)

Harletz
Ionnina • Soreq
ODP977 •

AI-1'
AI-2'
AI-3'
AI-4'
AI-5'-b
AI-5' AI-5'-a
AI-6'
AI-7'
AI-8'
AI-9'-b
AI-9' AI-9'-a
AI-10'

I1
I2
I3
I4
I5
I6
I7
I8
I9
I10
I11
I12

**Sapropel 6**
Bard et al., 2002

Roucoux et al., 2011
**Ionnina**
AP-(P+B+J) (%)
AP(%)

**Summer insolation gradient** (65°N-15°N) (W/m2)

**Harletz** MIS6

New Fig. 4

[Figure]

**Age** (kyr) — **GLτ_syn** δ¹⁸O (‰) — Barker et al., 2011

**ODP983** N. pachy. s. (%) — Barker et al., 2013

**ODP977-MD01-2443** Uk'37 SST (°C) — Martrat et al., 2004, 2007

ice sheet extent by F. Colleoni from Dendy et al., 2017

**Age** (kyr)

Mokeddem & McManus 2016 — **ODP984** N. pachy. s. (%)

Obrochta et al., 2014 — **U1308** N. pachy. s. (%)

Wang et al., 2008 — **Sanbao 11** δ¹⁸O (‰)

**Harletz** MIS6

Predicted D-O event occurrence in Barker et al., 2011

New Tab. 1

Predicted D-O event occurrence

| Age kyr (EDC3) | SpeloAge (kyr) | DO pick | DO pick variable threshold | Sanbao11 | ODP984 | ODP983 | U1308 | ODP977/ MD01-2443 | Ionnnina | Soreq | Harletz | MIS6 | Max/ Min | Nb | frequence (ka) |
|---|---|---|---|---|---|---|---|---|---|---|---|---|---|---|---|
| | | | | B01 | IS6-1 | | | AI-1' | | | | | | | |
| | | | | | | | | | I1 | S1 | | | | | |
| | | | | BO2 | IS6-2 | x | U1308-1 | | I2 | S2 | | | | | |
| | | | | | | | | AI-2' | | | | | | | |
| | | | | | | | | | | | | | | | |
| | | | | B07 | IS6-3b | | U1308-2 | | I3 | S3 | | | | | |
| | | | | | IS6-3 | | | | | | | | | | |
| 146,70 | 147,37 | 1 | 1 | B08 | IS6-a | | | | | | | MIS6 P2: 130-170 ka | max: | 15 | 2,67 |
| | | | | | | | U1308-3 | | | | | | | | |
| 148,30 | 149,08 | 0 | 1 | B09 | | x | | AI-3' | | S4 | | | | | |
| 150,06 | 150,97 | 1 | 1 | B10 | IS6-b | | U1308-4b | AI-4' | | S5 | | | min: | 4 | 10,00 |
| | | | | | IS6-4 | | | | | | | | | | |
| | | | | B11 | | | U1308-4 | AI-5'-b | I4 | | | | | | |
| 159,12 | 160,69 | 1 | 1 | B12 | | x | U1308-5 | AI-5'-a | I5 | | | | | | |
| 162,16 | 164,03 | 1 | 1 | B13 | IS6-5 | x | U1308-6b | | | S6 | | | | | |
| | | | | B14 | | | U1308-6 | AI-6' | I6 | S7 | | | | | |
| 168,16 | 169,48 | 1 | 1 | B15 | | x | | AI-7' | I7 | | | | | | |
| | | 5 | 6 | 15 | 8 | 5 | 8 | 8 | 7 | 7 | 5 | # DO P2 | | | |
| 172,00 | 173,18 | 1 | 1 | B16 | | | | | I8 | S8 | | | | | |
| 174,98 | 176,29 | 1 | 1 | B17 | | | U1308-7 | AI-8' | I9 | | | | | | |
| 177,14 | 177,96 | 1 | 1 | | | x | U1308-8 | | | | | MIS6 P1: 170-192ka | max: | 5 | 4,40 |
| 178,30 | 178,65 | | | | | | | | | | | | | | |
| | | | | B18 | | x | | AI-9'-b | I10 | S9 | | | | | |
| | | | | | | | | AI-9'-a | I11 | | | | min: | 2 | 11,00 |
| 187,84 | 189,98 | 1 | 1 | B19 | | x | U1308-9 | AI-10' | I12 | | | | | | |
| 192,16 | 192,68 | 1 | 1 | B20 | | x | x | x | | | | | | | |
| | | 5 | 5 | 5 | | | 4 | 4 | 5 | 5 | 2 | 5 | # DO P1 | | |
| | | | | | | | | | | | | | max: | 20 | 3,10 |
| | | 10 | 11 | 20 | 8 | 9 | 12 | 13 | 12 | 9 | 10 | # DO MIS6 | | | |
| | | | | | | | | | | | | | min: | 7 | 8,86 |